# Genomic and Reverse Translational Analysis Discloses a Role for Small GTPase RhoA Signaling in the Pathogenesis of Schizophrenia: Rho-Kinase as a Novel Drug Target

**DOI:** 10.3390/ijms242115623

**Published:** 2023-10-26

**Authors:** Rinako Tanaka, Kiyofumi Yamada

**Affiliations:** 1Department of Neuropsychopharmacology and Hospital Pharmacy, Graduate School of Medicine, Nagoya University, Nagoya 466-8560, Japan; rtanaka@med.nagoya-u.ac.jp; 2International Center for Brain Science (ICBS), Fujita Health University, Toyoake 470-1192, Japan

**Keywords:** copy number variants, CNVs, single-nucleotide polymorphisms, SNPs, fasudil, Kalirin, dopamine, glutamate, RTN4R, KCTD13

## Abstract

Schizophrenia is one of the most serious psychiatric disorders and is characterized by reductions in both brain volume and spine density in the frontal cortex. RhoA belongs to the RAS homolog (Rho) family and plays critical roles in neuronal development and structural plasticity via Rho-kinase. RhoA activity is regulated by GTPase-activating proteins (GAPs) and guanine nucleotide exchange factors (GEFs). Several variants in GAPs and GEFs associated with RhoA have been reported to be significantly associated with schizophrenia. Moreover, several mouse models carrying schizophrenia-associated gene variants involved in RhoA/Rho-kinase signaling have been developed. In this review, we summarize clinical evidence showing that variants in genes regulating RhoA activity are associated with schizophrenia. In the last half of the review, we discuss preclinical evidence indicating that RhoA/Rho-kinase is a potential therapeutic target of schizophrenia. In particular, Rho-kinase inhibitors exhibit anti-psychotic-like effects not only in *Arhgap10* S490P/NHEJ mice, but also in pharmacologic models of schizophrenia (methamphetamine- and MK-801-treated mice). Accordingly, we propose that Rho-kinase inhibitors may have antipsychotic effects and reduce cognitive deficits in schizophrenia despite the presence or absence of genetic variants in small GTPase signaling pathways.

## 1. Introduction

Schizophrenia is one of the most serious psychiatric disorders and affects approximately 1% of the population [1]. It typically emerges in late adolescence and early adulthood and involves positive symptoms (such as hallucinations, delusions, and formal thought disorder), negative symptoms (such as lack of volition, reduced speech output, and flattening of affect), and cognitive dysfunction (manifested, for instance, by deteriorations in working memory, executive function, and learning) [1,2,3].

Neuropathological and neurophysiological changes observed in patients with schizophrenia include enlargement of the lateral ventricles and a 2% decrease in gray matter volume [4]. Brain volume reduction involves the frontal lobe in particular, including the frontal cortex, which exhibits a reduced density of pyramidal neuron spines that are components of the postsynaptic site of most excitatory synapses [4,5,6,7,8,9]. Moreover, patients with schizophrenia show decreased prefrontal cortex (PFC) blood flow during the performance of cognitive tasks [10].

One of the major therapeutic targets of schizophrenia is the dopamine D2 receptor, and its antagonists, such as haloperidol, reduce positive symptoms but have a minimal effect on negative symptoms or cognitive deficits [1,11]. These drugs also have major side effects, including sedation, hyperprolactinemia, and the extrapyramidal symptoms of parkinsonism [1]. Second-generation antipsychotics such as risperidone and olanzapine have lower rates of such severe side effects, but their clinical efficacy and tolerability are not significantly improved [12]. Furthermore, 20–30% of patients show resistance to antipsychotic treatment [13]. Clozapine is the sole drug indicated for treatment-resistant schizophrenia and improves symptoms in only about 30–60% of patients [14,15,16,17]. Therefore, there is an urgent need for the development of effective schizophrenia treatments that are both effective and safe. To achieve this goal, it is necessary to understand the pathoetiology of the disease and to establish novel pathophysiologic animal models to expand upon existing classical pharmacologic animal models.

The etiology of schizophrenia involves both genetic vulnerabilities and environmental risk factors such as pregnancy and birth complications, childhood trauma, substance abuse, and psychosocial stress in adolescence [1,18]. In genome-wide association studies (GWASs) of schizophrenia, more than 200 genetic loci associated with neuronal function, including synaptic organization, differentiation, and transmission, have been shown to be associated with schizophrenia [19]. In addition to common variants, a small number of rare copy number variants (CNVs) and gene-disrupting variants, including the so-called rare-coding variants and protein-truncating variants, have been identified in schizophrenia with large effect sizes (odds ratios (ORs) of 2–60 fold and 3–50 fold, respectively) [1]. Several CNVs, such as 1q21.1, 2p16.3 (NRXN1), 3q29, 15q11.2, 15q13.3, and 22q11.2 have been consistently reported to be associated with schizophrenia [20,21]. Furthermore, a gene set analysis replicated previous findings (e.g., those implicating synapses and calcium signaling) and identified novel biological pathways such as those involved in the oxidative stress response, genomic integrity, and kinase and small GTPase signaling [22].

The Rho GTPase family plays a role in spine morphology by regulating actin dynamics [23]. It is associated with psychiatric diseases such as schizophrenia and depression, and also with neurodevelopmental disorders including autism spectrum disorders and intellectual disabilities [24,25,26,27,28,29,30,31]. Variants in genes upstream of the Rho GTPase family, such as *KALRN* and *p250GAP*, have been reported to be associated with schizophrenia [25,27]. However, few reviews have summarized schizophrenia-associated variants of genes regulating the Rho GTPase family. Here, we focus on RhoA, one of the Rho GTPases, and summarize its genetic association with schizophrenia and the effect of RhoA signaling modification in animal models of schizophrenia.

## 2. Rho Family Activity Is Regulated by GTPase-Activating Proteins (GAPs) and Guanine Nucleotide Exchange Factors (GEFs)

RhoA belongs to the RAS homolog (Rho) family, along with cell division control protein 42 (Cdc42) and RAS-related C3 botulinum toxin substrate 1 (Rac1) [32]. Rho family proteins contain a conserved GDP/GTP binding domain and switch their activity by cycling between GDP-bound (inactive) and GTP-bound (active) states [32,33]. This cycling is regulated by GAPs, GEFs, and guanine nucleotide dissociation inhibitors (GDIs) [32,34]. GAPs consist of more than 70 members, and conversion from a GTP-bound form to a GDP-bound form suppresses their activity [32,35]. In contrast, GEFs (>74 members) accelerate the exchange of tightly bound GDP for GTP, resulting in the activation of Rho family proteins [32,35]. GAPs and GEFs exhibit high selectivity for RhoA, Cdc42, and Rac1 [35,36]. GDIs, of which there are only three members in the human genome, form soluble complexes with GDP-bound Rho protein and control its cycling between the cytosol and membrane [32,34].

## 3. Rho Family Protein Effectors and Their Physiological Roles in the Brain

Rho family proteins are associated with over 70 potential effector proteins [37]. Rho-kinase, a serine/threonine kinase, is a representative downstream effector of RhoA [38]. In vascular smooth muscle, for example, Rho-kinase phosphorylates myosin phosphatase-targeting subunit 1 (MYPT1) at Thr696 and Thr853. This converts MYPT1 to an inactivated state, increases the phosphorylation of myosin light chain, and promotes actomyosin contractility [39,40,41]. P21-activated kinase (PAK) acts as a downstream effector for Cdc42 and Rac1 and affects actin dynamics by regulating the LIM kinase–cofilin pathway [42,43]. PAK also inhibits myosin light chain kinase, resulting in decreased myosin light chain phosphorylation and, thus, decreased actomyosin contractility [44].

Rho GTPases regulate cell morphology. For instance, RhoA promotes stress fiber formation and focal adhesions in cells [45]. Rho GTPases also modulate neuronal development. For instance, RhoA inhibits growth of dendrites and axons, while Rac1 and Cdc42 promote axonal elongation [42,46]. In addition, RhoA/Rho-kinase signaling promotes spine shrinkage and destabilization, while Rac1 and Cdc42/PAK signaling promotes spine stabilization and maintenance [23,47]. Accordingly, Rho GTPase signaling is involved in neuronal maturation through the regulation of actin dynamics.

## 4. Schizophrenia-Associated Genes Involved in Small GTPase RhoA Signaling

Recently, several variants of RhoA-associated GAPs, including *ARHGAP10* [48], *ARHGAP18* [49,50,51], and *p250GAP* [52], and GEFs such as *KALRN* [53,54,55,56] and *ARHGEF11* [57], were reported to be significantly associated with the development of schizophrenia. In addition, variants in genes that activate RhoA via GEF, such as *RTN4R* [58,59,60], or in those that degrade RhoA, such as *KCTD13* [61], were also identified in schizophrenia (Table 1, Figure 1).

### 4.1. GAPs

#### 4.1.1. ARHGAP10

The *ARHGAP10* gene, which encodes Rho GTPase-activating protein 10 (ARHGAP10), is located on chromosome 4q31.23 and exhibits GAP activity for RhoA and Cdc42 [32,68]. ARHGAP10 is expressed in the brain [68,69], and its mRNA levels rise in the cerebellum, striatum, and frontal cortex from E4 to P56 in mice [69].

CNVs in *ARHGAP10* were identified in seven patients with schizophrenia (six with deletions and one with duplication) but not in controls, and there was a significant association of *ARHGAP10* CNVs with schizophrenia in Japanese patients (OR = 12.3, *p* = 0.015) [48]. Most *ARHGAP10* CNVs were exonic deletions at the Bin1/amphiphysin/Rvs167 domain, the RhoGAP domain, or both. The relative expression levels of *ARHGAP10* mRNA in lymphoblastoid cell lines established from the peripheral blood of patients with exonic *ARHGAP10* CNVs were significantly decreased compared to those in patients with schizophrenia without *ARHGAP10* CNVs and in a control group [48]. One of the patients (case #5) with *ARHGAP10* CNVs had a missense variant (p.S490P) in exon 17 that overlapped with the exonic deletion on the other allele [48]. ARHGAP10 p.S490P showed weaker binding to active RhoA compared to wild-type ARHGAP10, suggesting that this single-nucleotide variation (SNV) exhibits the loss of function of ARHGAP10. Of note, clinical data of these seven patients with *ARHGAP10* variants showed that treatment response was poor in most individuals, including case #5 [48].

#### 4.1.2. ARHGAP18

ARHGAP18 is ubiquitously expressed throughout the body, including the brain, and shows GAP activity for RhoA but not for Rac1 or Cdc42 [70,71,72]. Through RhoA/Rho-kinase signaling, ARHGAP18 regulates cell spreading and migration and also the formation of stress fibers and focal adhesions [70]. ARHGAP18 knockdown, which leads to RhoA activation, causes significantly increased formation of stress fibers and focal adhesions in HeLa cells, while these changes are abolished by a Rho-kinase inhibitor or by dominant-negative RhoA transfection [70]. In contrast, the overexpression of wild-type ARHGAP18, but not GAP-defective ARHGAP18, suppresses the formation of stress fibers and focal adhesions in HeLa cells [70]. ARHGAP18 also contributes to cell migration [70]. The knockdown or knockout of *ARHGAP18* impairs migration and cellular polarity in breast cancer cells (MDA-MB-231 or SUN-159 cells) [70,72].

Single-nucleotide polymorphisms (SNPs) in *ARHGAP18* are associated with schizophrenia [49,50]. The genotypes and allelic frequencies of two SNPs, rs7758025 and rs9483050, were significantly different between patients and controls in a Chinese-Han population (genotype: rs7758025, *p* = 0.0002, and rs9483050, *p* = 7.54 × 10^−6^; allelic frequencies: rs7758025, *p* = 4.36 × 10^−5^, and rs9483050, *p* = 5.98 × 10^−7^). In addition, the AG haplotype in rs7758025-rs9385502 and the CG haplotype in rs11753915-9483050 were associated with schizophrenia (AG haplotype in rs7758025-rs9385502: *p* = 0.0012, 95% confidence interval [CI] = 0.48–0.93; CG haplotype in rs11753915-9483050: *p* = 9.6 × 10^−6^, 95% CI = 0.44–0.78) [49]. Another group reported an association between SNPs in *ARHGAP18* and schizophrenia in Caucasian people [50,51]. They also performed a combined analysis with GWAS and functional magnetic resonance imaging scanning and demonstrated that these SNPs were significantly correlated with neuronal activity in the dorsolateral prefrontal cortex during a working memory task [50].

#### 4.1.3. p250GAP (ARHGAP32)

P250GAP (ARHGAP32) is expressed in the brain [73] and exerts GAP activity for RhoA but not Cdc42 and Rac1 in mouse primary hippocampus neurons [74]. P250GAP regulates spine morphology in primary hippocampus neurons and neurogenesis in Neuro-2A cells through the regulation of RhoA activity [74,75]. In addition, p250GAP interacts with the NR2B subunit of N-methyl-D-aspartate (NMDA) receptors and is involved in NMDA receptor-mediated RhoA activation [74,75].

An SNP in *p250GAP* (rs2298599) was shown to be associated with schizophrenia in a Japanese cohort (*p* = 0.00015) [52]. The minor genotype frequency was higher in patients with schizophrenia (18%) than in healthy controls (9%) (*p* = 0.000083) [52]. rs2298599 is located 2.9 kb downstream of p250GAP and showed no significant association with p250GAP expression levels in immortalized lymphoblasts in in silico analysis (*p* = 0.28) [52]. Thus, further study is needed to clarify the mechanism of the association between p250GAP and schizophrenia.

### 4.2. GEFs

#### 4.2.1. *KALRN*

Kalirin has two GEF domains, with activity targeting Rac1 (GEF1) and RhoA (GEF2), respectively [27,76]. The alternative splicing of *KALRN* gives rise to several isoforms, including Kalirin-4, Kalirin-5, Kalirin-7, Kalirin-8, Kalirin-9 (Kal9), and Kalirin-12 (Kal12) [27,76]. Kal9 and Kal12 contain both the GEF1 and GEF2 domains, while other isoforms contain only the GEF1 domain [27,76]. Kalirin is involved in neurite and dendritic outgrowth and in dendritic arborization in the brain [76]. Expression levels of Kal9 and Kal12 in the brain were found to be higher during early postnatal development than in adulthood [77]. Knockout of Kal9 and Kal12 by shRNA decreased the complexity of rat hippocampal neurons on days in vitro (DIV) 4 and 7 (immature neurons) [78]. In rat cortical neurons, Kal9 overexpression on DIV 2 (immature neurons) resulted in neurite elongation [77], while that on DIV 28 (mature neurons) reduced dendritic length and complexity [79]. These reports indicate that the role of Kal9 in neurons might change depending on the developmental stage.

A transcriptome-wide association study in patients with schizophrenia revealed an increase in exon skipping immediately prior to the GEF2 domain in *KALRN* transcripts [53]. In addition, some missense variants in *KALRN*, such as P2255T and T1207M, showed a higher frequency in schizophrenia cases compared to control cases [54,56]. In particular, P2255T in *KALRN* was significantly associated with schizophrenia (OR = 2.09, *p* = 0.048) in a Japanese population [54]. Of note, a P2255 residue exists near the RhoA-GEF2 domain [55]. The P2255T variant in Kal9 (Kal9-P2255T) leads to highly stable Kal9 mRNA, resulting in increased protein levels of Kal9 [55], which, for instance, were detected in the auditory cortex of patients with schizophrenia by post-mortem analysis [79]. Furthermore, overexpression of Kal9-P2255T in rat primary neurons and HEK 293 cells increased RhoA activity but had no effect on Rac1 activity [55,80]. These data indicate that Kal9-P2255T increases the expression levels of Kal9, leading to the activation of RhoA. From the viewpoint of neuronal morphology, Kal9-P2255T overexpression in cortical primary neurons led to a significant reduction in proximal dendritic complexity and dendritic spine size compared to wild-type Kal9 (Kal9-WT) [55]. On the other hand, it is known that reticulon 4 receptor (RTN4R) activates Kal9 and subsequently leads to the activation of RhoA [63,81]. The RTN4R/Kal9/RhoA pathway is known to modulate neurite outgrowth [81]. Myelin-associated inhibitors such as oligodendrocyte-myelin glycoprotein (OMGp) have been identified as additional RTN4R (NGR1) ligands and these also suppress neurite outgrowth [80,82]. Pharmacologic inhibition of RhoA with the RhoA inhibitor CT04 prevented the OMGp-induced decrease in neuronal complexity [80]. The overexpression of Kal9-P2255T in cortical primary neurons made them more sensitive to OMGp and decreased both the length and complexity of dendritic arbors [80]. These results suggest that Kal9-P2255T-induced RhoA activation causes morphological changes in neurons.

#### 4.2.2. ARHGEF11

ARHGEF11, also referred to as KIAA0380 or GTRAP48, shows GEF activity for RhoA but not Rac1 or Cdc42 [83]. ARHGEF11 is expressed in the brain [84,85] and in cortical neurons, including dendrites and spines [86]. ARHGEF11 regulates glutamate transport activity by direct binding of excitatory amino acid transporter 4 [85]. The overexpression of ARHGEF11 was shown to decrease spine density in rat cortical neurons [86,87].

*ARHGEF11* haplotypes such as C-C of rs6427340-rs6427339 and A-C-C of rs822585-rs6427340-rs6427339 were shown to be associated with schizophrenia (*p* = 0.0010 and 0.0018, respectively), but the *ARHGEF11* SNPs were not [57]. The functions of *ARHGEF11* haplotypes associated with schizophrenia have not been clarified. On the other hand, in situ hybridization analysis indicated that *ARHGEF11* mRNA levels in the thalamus of patients with schizophrenia were higher than those in healthy controls [88]. These findings raise the possibility that *ARHGEF11* activation is associated with schizophrenia pathology.

### 4.3. Others

#### 4.3.1. *RTN4R*

RTN4R (also called Nogo-66 receptor, NgR1) is a RTN4 receptor subunit located at chr22q11.2, and it has been shown that deletion of chr22q11.2 is associated with a high risk of developing schizophrenia [89]. RTN4R binds leucine-rich repeat and immunoglobulin domain-containing protein (Lingo-1) and either the p75 neurotrophin receptor or tumor necrosis factor (TNF) receptor orphan Y (TROY) and activates RhoA through Kal9; this results in the collapse of growth cones, which prevents further axonal growth and inhibits myelination [62,63].

Several SNPs in RTN4R are associated with schizophrenia [58,59]. In samples from individuals of Afrikaner origin, significant associations with schizophrenia were seen for SNP rs696880 in women (OR = 0.73, *p* = 0.046) and for rs701427 (OR = 1.21, *p* = 0.019), rs696880 (OR = 1.18, *p* = 0.029), and rs854971 (OR = 1.20, *p* = 0.021) in men [59]. Diffusion tensor imaging revealed that the SNP rs701428 was associated with white matter abnormalities in 22q11.2 deletion syndrome [90]. Other groups identified several rare missense variants in patients with schizophrenia, specifically p.R68H (rs145773589), p.R119W (rs74315508), p.R196H (rs74315509), p.D259N (rs3747073), p.R292H (rs1432033565), and p.V363M (rs149231717), and p.R292H was significantly associated with schizophrenia (OR = 3.9, *p* = 0.048) [58,60]. RTN4R-R292H is located in the ligand binding site, and its overexpression in E5.5 chick retinal neurons significantly decreased growth cone collapse induced by treatment with RTN4, a ligand of RTN4R, compared to that resulting from treatment with RTN4R-WT [58]. In addition, a glutathione S-transferase binding assay showed that RTN4R-R292H exhibited reduced interaction with LINGO1 compared to RTN4R-WT [58]. Although these data suggest that RTN4R-R292H has impaired function, the effect of RTN4R-R292H on RhoA signaling remains obscure. In a post-mortem brain analysis, the expression levels of RTN4R were decreased in the dorsolateral prefrontal cortex but increased in the hippocampal CA3 region of patients with schizophrenia compared to healthy controls [91].

In addition to the above, genetic variations in components of the RTN4R signaling pathway, such as RTN4 (*p* = 0.047 and 0.037 for rs11894868 and rs2968804, respectively) and myelin-associated glycoprotein (*p* = 0.034 and 0.029 for rs7249617 and rs16970218, respectively), were shown to be associated with schizophrenia [92].

#### 4.3.2. 16p11.2 CNVs and the KCTD13-Cul3-RhoA Pathway

16p11.2 microduplication was associated with schizophrenia in two large cohorts (OR = 25.8, *p* = 1.2 × 10^−5^; and OR = 8.3, *p* = 0.022) [61]. In a zebrafish model, potassium channel tetramerization domain-containing 13 (KCTD13) was identified as the sole signaling protein capable of inducing the microcephaly phenotype associated with 16p11.2 duplication [64]. A spatiotemporal protein–protein interaction network analysis showed that KCTD13 is functionally related to cullin 3 (Cul3) [67]. Cul3 is a core component of E3 ubiquitin–protein ligase complexes and mediates the ubiquitination and subsequent proteasomal degradation of target proteins such as RhoA, but not Rac1 and Cdc42 [65,66]. Although it is estimated that 16p11.2 microduplication is associated with decreased RhoA protein levels, organoids derived from patients with autism spectrum disorder who had 16p11.2 duplications showed RhoA activation and slightly increased KCTD13 expression [93]. Therefore, further research should analyze 16p11.2 microduplication in patients with schizophrenia.

## 5. Crosstalk between Ras and Rho Signaling in Schizophrenia

In cancer, p120RasGAP inhibits the RhoGAP activity of Deleted in liver cancer 1 (DLC1, i.e., STARD12, ARHGAP7), which is a tumor suppressor [94]. Crossveinless-c, the Drosophila homolog of DLC1, regulates the elongation of dendritic branches via RhoA activity [95]. On the other hand, integrin-mediated activation of Abl2/Arg and Src-family kinases increases p190RhoGAP phosphorylation to drive its association with p120RasGAP at the cell membrane, resulting in inhibition of RhoA activity and stabilizing the dendrite structure [96]. However, there are few reports about the crosstalk between Ras and Rho signaling, including *p120RasGAP*, *DLC1*, and *p190RhoGAP* gene variants, in schizophrenia. Only one group reported the association of p190RhoGAP with the pathway of acid phosphatase 1 (ACP1), which was associated with suicide attempts in Caucasians with primary diagnoses of schizophrenia and schizoaffective disorder [97]. Thus, further studies are required to discuss the crosstalk between Ras and Rho signaling in schizophrenia.

## 6. Genetic Mouse Models of Schizophrenia with Associated Genetic Variants Involved in Small GTPase RhoA Signaling

Mouse models have been developed based on schizophrenia-associated genetic variants involved in small GTPase RhoA signaling, including variants affecting the *Arhgap10*, *Kalrn*, and *Rtn4r* genes, and their phenotypic characterization has been performed [48,55,59,69,80,98] (Table 2).

### 6.1. Arhgap10 S490P/NHEJ Mice

*Arhgap10* S490P/NHEJ mice carry double variants of the *Arhgap10* gene that mimic the *ARHGAP10* variations discovered in a Japanese patient with schizophrenia (case #5). One allele contains a missense variant (p.S490P), while the other contains a frameshift variant caused by non-homologous end joining (NHEJ) [48]. Compared to wild-type littermates, these mice exhibit significantly increased levels of both phosphorylated MYPT1 at Thr696 in the medial PFC (mPFC), striatum, and nucleus accumbens (NAc), and of phosphorylated p21-activated kinase (PAK) (PAK1 at Ser144 and PAK2 at Ser141) in the striatum and NAc. These results suggest that Rho family RhoA and Cdc42 signaling is abnormally activated in the mPFC, striatum, and NAc of *Arhgap10* S490P/NHEJ mice [69].

A neuropathological analysis showed that spine density in *Arhgap10* S490P/NHEJ mice was decreased in the mPFC but increased in the striatum [48,69]. Of note, repeated oral administration of fasudil, a Rho-kinase inhibitor, at a dose of 20 mg/kg for 7 days rescued the decreased spine density in the mPFC in *Arhgap10* S490P/NHEJ mice but had no effect in wild-type mice [99]. These results suggest that abnormal activation of RhoA/Rho-kinase signaling in *Arhgap10* S490P/NHEJ mice causes a reduction in spine density in the mPFC. Furthermore, the group that performed the aforementioned study established induced pluripotent stem cells (iPSCs) from patients with schizophrenia and differentiated them into tyrosine hydroxylase (TH)-positive neurons in order to analyze their morphological phenotypes. The TH-positive neurons differentiated from the iPSCs of the patient identified as case #5 exhibited severe defects in both neurite length and branch number, which were restored by the addition of the Rho-kinase inhibitor Y-27632 [48]. These findings suggest that Rho-kinase plays significant roles in the neuropathological changes in spine morphology caused by *ARHGAP10* variants (Table 3).

Comprehensive behavioral analyses revealed increased anxiety and vulnerability to methamphetamine-induced impairment in locomotion and cognitive function in *Arhgap10* S490P/NHEJ mice [48,69]. This phenotype is consistent with evidence that psychostimulants, including amphetamine and methamphetamine, cause psychotic symptoms and cognitive dysfunction in patients with schizophrenia at doses that show little effect in healthy controls [104,105,106]. Notably, acute treatment with fasudil rescued the methamphetamine (0.3 mg/kg, i.p.)-induced cognitive impairment in the visual discrimination tasks in *Arhgap10* S490P/NHEJ mice [99]. Fasudil also suppressed c-Fos expression in the mPFC that was induced by low-dose methamphetamine in *Arhgap10* S490P/NHEJ mice [99] (Table 3).

Collectively, these results suggest that schizophrenia-associated *Arhgap10* gene variants result in morphological abnormalities of neurons in the mPFC, and these abnormalities are associated with vulnerability to cognitive deficits induced by methamphetamine treatment. *Arhgap10* S490P/NHEJ mice are a unique genetic mouse model of schizophrenia with constructive, phenotypic, and predictive validity.

### 6.2. Kalrn P2255T Mice

*Kalrn* P2255T mice harbor the missense variant P2255T at the endogenous locus in *Kalrn*. In these mice, impaired prepulse inhibition (PPI) is caused by various intervals between prepulse and startle-eliciting noise (Gap-PPI), but not by various noises at a lower sound pressure level than the startle-eliciting noise (Noise-PPI) [80]. Noise-PPI depends on subcortical auditory processing [107], while Gap-PPI has been shown to require the primary auditory cortex [108]. These findings suggest that P2255T in *Kalrn* affects primary auditory cortex functioning but not subcortical auditory processing. Indeed, *Kalrn* P2255T mice showed reductions in both dendritic length and the complexity of layer 3 pyramidal neurons in primary auditory neurons at 12 weeks old [80]. These results suggest that *Kalrn* P2255T mice constitute a genetic model that reflects schizophrenia pathology. Further research should investigate RhoA activity in the brains of these mice and the effect of RhoA/Rho-kinase inhibition on their phenotypes in order to clarify the pathomechanism underlying the *Kalrn* P2255T variant.

### 6.3. Rtn4r Knockout Mice

One research group generated *Rtn4r* knockout mice in which exon 2 in the *Rtn4r* gene was deleted, and therefore RTN4R expression was selectively abolished [109]. These mice showed delayed learning of a spatial memory task in a water maze test [98]. Another group reported a decrease in both distance traveled and rearing activity in an open-field test in different *Rtn4r* knockout mice [59]. Because schizophrenia is associated with deletion of 22q11.2 [89], where RTN4R is located, these reports suggest that RTN4R expression may contribute to the etiology of schizophrenia.

## 7. RhoA/Rho-Kinase Signaling in a Pharmacological Model of Schizophrenia

So far, we have reviewed genomic/genetic and reverse translational studies that suggest a role of RhoA signaling in the pathogenesis of schizophrenia. Furthermore, potential antipsychotic-like effects of RhoA/Rho-kinase inhibitors have been demonstrated in genetic mouse models harboring schizophrenia-associated variants of genes related to RhoA signaling [48,49,50,51,52,54,56,57,58,59,60,61,99]. In particular, rare *ARHGAP10* variants are genetically and biologically associated with schizophrenia, and Rho-kinase may represent a promising drug target for schizophrenia treatment in patients with variants of *ARHGAP10* and possibly other genes related to the RhoA/Rho-kinase pathway [99].

In terms of drug development, it is important to assess whether Rho-kinase inhibitors exhibit antipsychotic effects in patients with schizophrenia who carry no variants in *ARHGAP10* or related genes. In the field of pain therapy, inhibitors of the voltage-gated sodium channel Nav1.7 are currently in phase II/III clinical trials (NCT02935608 and NCT02365636) for the treatment of chronic pain [110]. A genomic analysis identified a missense variant in *SCN9A*, which encodes the α-subunit of the voltage-gated sodium channel Nav1.7 in a Chinese family with a rare autosomal dominant form of erythromelalgia, which led to the discovery of this drug target [110]. Nav1.7 inhibitors were developed and their effectiveness has been evaluated in animal models of pain without *SCN9A* variants (i.e., wild-type) [111]. Based on these evidences, Nav1.7 blockers are being developed as novel treatments for chronic pain, regardless of the presence or absence of *SCN9A* variants.

From the viewpoint of drug development, we have reported the effects of Rho-kinase inhibitors in pharmacological models of schizophrenia that lack variants of RhoA-related genes (Table 3).

### 7.1. Dopamine Hypothesis-Based Model (Methamphetamine Treatment Model)

A disturbance of dopamine function is considered to be one of the primary factors underlying schizophrenia (i.e., the dopamine hypothesis) [1]. Methamphetamine and amphetamine are widely used to induce schizophrenia-like behavior in rodents [1,112,113], and there is evidence that both activate RhoA activity [100,114,115,116,117]. Acute methamphetamine treatment (1 or 2 mg/kg, i.p.) was shown to increase the phosphorylation levels of MYPT1 and myosin light chain 2, both of which are substrates of Rho-kinase, in the mPFC and dorsomedial striatum in wild-type mice [100]. In addition, in acute slices of mouse midbrain, as well as in the neuroblastoma cell line SK-*N*-SH, RhoA and Rac were activated 5 min after amphetamine treatment (10 μM) but their total expression levels were not altered [114]. On the other hand, chronic methamphetamine treatment at a neurotoxic dose increased RhoA and Rho-kinase expression levels in rat hippocampi (15 mg/kg, i.p., eight times at 12 h intervals) [115], rat brain microvascular endothelial cells (1.5 mM for 6 h or 10 nM for 24 h) [115,116], and PC12 cells (0.5–2.5 mM for 24 h) [117]. The mechanism by which methamphetamine and amphetamine activate RhoA remains unclear. It is known that both drugs increase cAMP levels and activate protein kinase A (PKA) [114,118]. PKA phosphorylates RhoA at Ser188, leading to inactivation of RhoA by enhancing its binding affinity with Rho GDP dissociation inhibitor [118,119,120]. Indeed, D1/D5 Gs-coupling receptor agonist SKF38393 suppressed amphetamine-induced RhoA activation in acute slices of mouse midbrain [114]. Therefore, it is unlikely that PKA activation by D1/D5 receptors is associated with amphetamine- or methamphetamine-induced RhoA activation.

There is some evidence that Rho-kinase inhibitors rescue methamphetamine-induced abnormal behaviors. Acute systemic treatment with fasudil (10–20 mg/kg, i.p. or 20 mg/kg, p.o.) rescued methamphetamine (1 mg/kg, i.p.)-induced cognitive impairment in visual discrimination tasks in wild-type mice [100]. NAc pretreatment with the Rho-kinase inhibitor Y-27632 suppressed methamphetamine-induced behaviors such as rearing and sniffing in rats [101]. One possible mechanism of Rho-kinase inhibitors is the inhibition of methamphetamine-induced dopamine elevation in the NAc. Pretreatment with Y-27632 in the NAc of rats suppressed the methamphetamine (1 mg/kg, subcutaneously)-induced elevation of extracellular dopamine levels, but had no effect on the methamphetamine-induced decrease in two major dopamine metabolites, 3,4-dihydroxyphenylacetic acid and homovanillic acid [101]. Another group reported that RhoA mediated the amphetamine-induced internalization of the dopamine transporter and suppressed dopamine uptake in acute slices of mouse midbrain and in SK-*N*-SH cells [114]. These results are consistent with the previous finding that pre-treatment with Y-27632 in the NAc of rats had no effect on the increase in extracellular dopamine levels induced by treatment with either tetrodotoxin, an inhibitor of voltage-dependent Na^+^ channels, or GBR-12909, a dopamine re-uptake inhibitor [101]. Alternatively, Rho-kinase inhibitors may suppress methamphetamine-induced neuronal activation. It was reported that fasudil (20 mg/kg, i.p.) suppressed methamphetamine (1 mg/kg, i.p.)-induced c-Fos expression in the mPFC and dorsomedial striatum [100]. Thus, this evidence suggests that RhoA/Rho-kinase signaling is abnormally activated under conditions in which dopaminergic neuronal activity is increased (i.e., schizophrenia). Accordingly, Rho-kinase is a potential novel target of drug discovery and development in schizophrenia.

### 7.2. Glutamate Hypothesis-Based Model (MK-801 Treatment Model)

Antagonists of NMDA receptors have also been shown to induce schizophrenia-like behaviors [1,121]. Previous studies reported that blocking NMDA receptor signaling by treatment with MK-801 or ketamine results in increased RhoA activity [122,123]. At 24–48 h after repeated MK-801 treatment (0.2 mg/kg/day, i.p.) for 14 days, adolescent rats showed impaired spatial memory and a lower proportion of mature spines, and also, in the hippocampus, increased mRNA levels of RhoA and decreased mRNA levels of Rac1 and Cdc42 [122]. MK-801 treatment of B35 neuronal cells and C6 glial cells (25 μM for 14 days) increased RhoA expression and myosin light chain 2 phosphorylation but decreased Cdc42 expression and PAK1 phosphorylation [123]. Another group reported that ketamine treatment (300 μM for 6 h) of rat hippocampal neurons on DIV 5 increased RhoA and Rho-kinase expression [103]. Treatment with Y-27632 rescued the ketamine-induced decrease in spine density in rat hippocampal neurons on DIV5 [103]. These data suggest that inhibition of NMDA receptor activity in neurons activates RhoA/Rho-kinase signaling, leading to decreased spine density.

Rho-kinase inhibition was also shown to rescue MK-801-induced abnormal behaviors in mice [102]. Fasudil rescued several MK-801-induced conditions, including hyperlocomotion (fasudil dose: 10–20 mg/kg, i.p.), social interaction impairment (10 mg/kg, i.p.), novel object recognition impairment (10–20 mg/kg, i.p.), and PPI deficits (20 mg/kg, i.p.) [102]. These data suggest that RhoA/Rho-kinase signaling contributes to schizophrenia-like behaviors in an MK-801 treatment model in animals.

## 8. Perspectives

As discussed in this review, RhoA/Rho-kinase is a potential therapeutic target in schizophrenia. In particular, Rho-kinase inhibitors exhibited anti-psychotic-like effects not only in *Arhgap10* S490P/NHEJ mice but also in pharmacologic models of schizophrenia (methamphetamine- and MK-801-treated mice). It is therefore expected that regardless of the presence or absence of genetic variants in the small GTPase signaling pathway, these inhibitors will have antipsychotic effects, in addition to their ability to ameliorate cognitive deficits in schizophrenia.

In general, optimizing drug target selection is an important step in drug discovery and development. Rho-kinase has many downstream molecules and plays various roles in the body [124]. Thus, Rho-kinase inhibitors may have unwanted side effects in clinical use. In this regard, there are two isoforms of Rho-kinase, namely Rho-kinase 1 and Rho-kinase 2. Rho-kinase 1 is expressed mainly in the lungs, liver, testis, blood, and immune system, while Rho-kinase 2 is found primarily in the brain, heart, and smooth muscle cells [124,125,126]. In the brain, Rho-kinase 1 is expressed in glial cells, whereas Rho-kinase 2 is expressed in neurons [126]. Thus, selective inhibition of Rho-kinase 2 may prevent unwanted peripheral side effects such as reduced blood pressure. Fasudil and Y-27632 are both dual inhibitors of Rho-kinase 1 and Rho-kinase 2. Selective inhibitors of Rho-kinase 2, such as KD025, have been developed [127,128], and it will be valuable to examine whether they have antipsychotic-like effects resembling those of fasudil, but with fewer side effects. Another potential means of avoiding unwanted side effects is to discover molecules downstream of Rho-kinase that are selectively expressed in the brain, especially in neurons. Therefore, the detailed mechanism of RhoA/Rho-kinase signaling in schizophrenia should be further clarified to facilitate the development of safe and effective therapeutic drugs for this disorder.

## Figures and Tables

**Figure 1 ijms-24-15623-f001:**
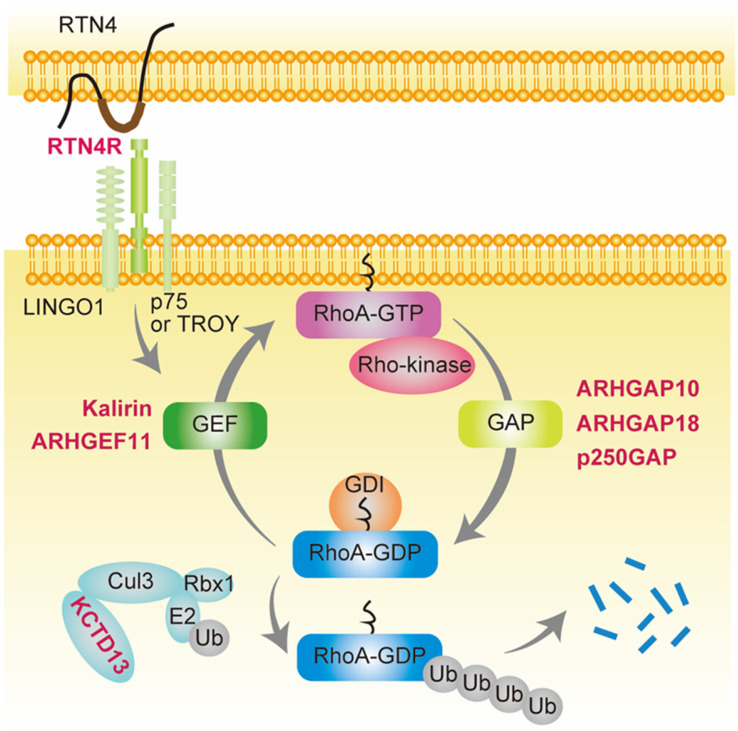
Schizophrenia-associated genes involved in small GTPase RhoA signaling. Genes shown in red are schizophrenia-associated genes involved in small GTPase RhoA signaling. RhoA contains a conserved GDP/GTP binding domain, and its activity cycles between GDP-bound (inactive) and GTP-bound (active) states [32,33]. ARHGAP10, ARHGAP18, and p250GAP are GTPase-activating proteins (GAPs) that convert RhoA from the GTP- to GDP-bound form, thereby suppressing its activity. In contrast, Kalirin and ARHGEF11 are guanine nucleotide exchange factors (GEFs) that accelerate the exchange of tightly bound GDP for GTP, resulting in RhoA activation. GDIs form soluble complexes with GDP-bound RhoA and control its cycling between the cytosol and membrane [32,34,35]. Reticulon 4 receptor (RTN4R) (also called Nogo-66 receptor, NgR1), a RTN4 receptor subunit, is activated by RTN4 and binds leucine-rich repeat and immunoglobulin domain-containing protein (Lingo-1) and either the p75 neurotrophin receptor or tumor necrosis factor (TNF) receptor orphan Y (TROY), resulting in RhoA activation by GEF [62,63]. Potassium channel tetramerization domain-containing 13 (KCTD13) is the only identified signaling protein capable of inducing the microcephaly phenotype associated with 16p11.2 duplication, which is associated with schizophrenia [61,64]. KCTD13 is functionally related to cullin 3 (Cul3), a core component of E3 ubiquitin-protein ligase complexes that mediates the ubiquitination and subsequent proteasomal degradation of target proteins such as RhoA [65,66,67].

**Table 1 ijms-24-15623-t001:** Genetic variants of RhoA GAPs/GEFs in schizophrenia.

Gene	Chromosomal Location	Function	Types of Variants and Changes ^a^	Functional Changes ^b^	Reference
*ARHGAP10*	4q31.23	GAP for RhoA and Cdc42	CNVs ↑ (deletion and duplication)	Loss of function	[48]
*ARHGAP18*	6q22.33	GAP for RhoA	SNPs ↑ (rs7758025 and rs9483050), Haplotypes ↑ (AG in rs7758025-rs9385502, CG in rs11753915-9483050)	ND	[49,50,51]
*p250GAP*	11q24.3	GAP for RhoA	SNP ↑ (rs2298599)	ND	[52]
*KALRN*	3q21.1-q21.2	GEF for Rac1 and RhoA	Exon skipping in transcriptome ↑, SNPs ↑ (P2255T and T1207M)	P2255T: leads to stable Kal9 mRNA	[53,54,55,56]
*ARHGEF11*	1q23.1	GEF for RhoA	Haplotypes ↑ (CC of rs6427340-rs6427339 and ACC of rs822585-rs6427340-rs6427339)	ND	[57]
*RTN4R*	22q11.21	Activation of GEF	SNPs ↓↑ (rs696880 (↑ male, ↓ female), rs701427 (↑ male), rs854971 (↑ male) and p.R292H (i.e., rs1432033565) ↑	R292H: loss of function	[58,59,60]
*16p11.2 (KCTD13* */Cul3)*	16p11.2	Degradation of RhoA	Microduplication ↑	ND	[61]

^a^ ↑ and ↓ indicate an increase or decrease in patients with schizophrenia compared to healthy control, respectively. ^b^ ND: Not determined.

**Table 2 ijms-24-15623-t002:** Animal models with schizophrenia-associated gene variants in genes related to RhoA GAPs/GEFs.

Genetic Mice Model	Rho GTPase Activity ^a^	Behavioral Phenotype	Neuronal Morphological Phenotype ^a^	Reference
*Arhgap10* S490P/NHEJ mice	Increased RhoA and Cdc42 activity in the mPFC, striatum, and NAc	Increased anxiety and vulnerability to methamphetamine-induced impairment in locomotion and cognitive function	Decreased spine density in mPFC Increased spine density in striatum	[48,69]
*Kalrn* P2255T mice	Increased RhoA activity but no changes in Rac1 activity after overexpression of Kal9-P2255T in rat primary neurons	Impaired prepulse inhibition by various gap durations between prepulse and startle-eliciting noise (Gap-PPI)	Decreased dendritic length and complexity of layer 3 pyramidal neurons in primary auditory neurons	[55,80]
*Rtn4r* knockout mice	ND	Delayed learning of spatial memory task in water maze test Decreased distance traveled and number of rears in open-field test	ND	[59,98]

^a^ ND: Not determined.

**Table 3 ijms-24-15623-t003:** Effects of Rho-kinase inhibitors in genetic and pharmacologic models of schizophrenia.

Model	Sample	Phenotype	Effective Dose of Rho-Kinase Inhibitor	Reference
Genetic model	*Arhgap10* S490P/NHEJ mice	Increased vulnerability to methamphetamine-induced cognitive function	Fasudil (3–20 mg/kg, i.p.)	[99]
		Decreased spine density in mPFC	Fasudil (20 mg/kg for 7 days, p.o.)	[99]
	TH-positive neurons differentiated from iPSCs that were established from a schizophrenia patient with *ARHGAP10* variants	Decreased neurite length and branch number	Y-27632 (1–10 μM for 12–60 h)	[48]
Dopamine hypothesis-based model	C57BL/6J mice	Methamphetamine (1 mg/kg, i.p.)-induced cognitive impairment in visual discrimination tasks	Fasudil (10–20 mg/kg, i.p. or 20 mg/kg, p.o.)	[100]
	Sprague Dawley rats	Methamphetamine (1 mg/kg, subcutaneously)-induced increase in extracellular dopamine levels	Pre-treatment with Rho-kinase inhibitor Y-27632 (1–10 nmol) in the NAc	[101]
Glutamate hypothesis-based model	C57BL/6J mice	MK-801 (0.3 mg/kg, i.p.)-induced hyperlocomotion	Fasudil (10–20 mg/kg, i.p.)	[102]
		MK-801 (0.1 mg/kg, i.p.)-induced deficits in social interaction	Fasudil (10 mg/kg, i.p.)	[102]
		MK-801 (0.1 mg/kg, i.p.)-induced deficits in novel object recognition	Fasudil (10–20 mg/kg, i.p.)	[102]
		MK-801 (0.2 mg/kg, i.p.)-induced deficits in PPI	Fasudil (20 mg/kg, i.p.)	[102]
	Primary hippocampal neurons from postnatal Sprague Dawley rats	Ketamine (300 μM)-induced decrease in spine density in rat hippocampal neurons	Y-27632 (10 μM for 6 h)	[103]

## Data Availability

Not applicable.

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
