# Peer review of "Genomic and Reverse Translational Analysis Discloses a Role for Small GTPase RhoA Signaling in the Pathogenesis of Schizophrenia: Rho-Kinase as a Novel Drug Target"

_ijms, 2023, doi:10.3390/ijms242115623_

Round 1

Reviewer 1 Report

In this review authors discuss mutated genes involved in the regulation of Rho GTPases that are associated with schizophrenia. By applying the concept of reverse translational approaches in transgenic animals an emerging role of Rho GTPases in schizophrenia is proposed. The data are complemented by discussing RhoA/Rho-kinase signalling in pharmacological models of schizophrenia and the authors have contributed original work by themselves in this field, recently.

 Rho GTPase belong to the Ras homolog family of proteins oscillating between the inactive GDP-bound and the signalling active GTP-bound conformation. While activation of Rho A/Rho- kinases results in axonal growth cone collapse and neuronal cell death, the activation of other Rho family GTPases, Cdc42 or Rac are stimulating dendritic growth, spine formation, and dendritic branching.

 Here the regulation of activities of RhoA and its opposing GTPases, Cdc42 and Rac are elaborated in detail considering mutations in guanine nucleotide exchange factors (GEFs), GTPase-activating proteins (GAPs) and guanine nucleotide dissociation inhibitors (GDIs) in the context of schizophrenia associated genes. Part of these genes are then discussed as investigations in the corresponding transgenic mouse models.

 Finally, pharmacological models of schizophrenia are discussed to implement precision medicine in schizophrenia. In the perspectives section the development of specific inhibitors of RhoA downstream effectors is proposed. Rho kinase A2 is specifically expressed in the brain neurons while Rho kinase 1 is expressed in glial cells. Specific inhibitors such as KD025 targeting neuronal Rho kinase A2 should be developed to avoid side effects by targeting glial Rho kinase A1 activity.

Minor points:

 Table 3 is difficult to read and should be re-edited: It is difficult to find out for the reader which model belongs to the respective phenotype.

 The cross talk between Ras- and Rho signalling could be discussed as well in the possible context of schizophrenia?

Author Response

The authors would like to thank the reviewers and editors for their time and effort in reviewing this manuscript. Please find the detailed responses below and the corresponding revisions highlighted changes in the re-submitted files.

Comment 1) Table 3 is difficult to read and should be re-edited: It is difficult to find out for the reader which model belongs to the respective phenotype.

Answer) We have revised Table 3 to easily find out for the reader which model belongs to the respective phenotype (Page 10).

 Comment 2) The cross talk between Ras- and Rho signalling could be discussed as well in the possible context of schizophrenia?

Answer) We have included the new section as follows.

  1. Crosstalk between Ras- and Rho-signalling in schizophrenia (Page 7, line 297-309)

In cancer, p120RasGAP inhibits RhoGAP activity of Deleted in liver cancer 1 (DLC1 i.e. STARD12, ARHGAP7), a tumor suppressor [1]. Crossveinless-c, the Drosophila homolog of DLC1, regulates elongation of dendritic branches via RhoA activity [2]. On the other hand, integrin-mediated activation of Abl2/Arg and Src-family kinases increases p190RhoGAP phosphorylation to drive its association with p120RasGAP at the cell membrane resulting in inhibition of Rho activity and stabilize dendrite structure [3]. However, there are few reports about the crosstalk between Ras- and Rho-signalling, including p120RasGAP, DLC1, and p190RhoGAP gene variants, in schizophrenia. Only one group reported the association of p190RhoGAP with the pathway of acid phosphatase 1 (ACP1) which have associated with suicide attempts in Caucasians with primary diagnoses of schizophrenia and schizoaffective disorder [4]. Thus, further studies are required to discuss the crosstalk between Ras- and Rho-signalling in schizophrenia.   

[1] J.E. Chau, K.J. Vish, T.J. Boggon, A.L. Stiegler, SH3 domain regulation of RhoGAP activity: Crosstalk between p120RasGAP and DLC1 RhoGAP, Nat Commun 13(1) (2022) 4788.

[2] D. Sato, K. Sugimura, D. Satoh, T. Uemura, Crossveinless-c, the Drosophila homolog of tumor suppressor DLC1, regulates directional elongation of dendritic branches via down-regulating Rho1 activity, Genes Cells 15(5) (2010) 485-500.

[3] L.P. Shapiro, R.G. Parsons, A.J. Koleske, S.L. Gourley, Differential expression of cytoskeletal regulatory factors in the adolescent prefrontal cortex: Implications for cortical development, J Neurosci Res 95(5) (2017) 1123-1143.

[4] J. Li, A. Yoshikawa, H.Y. Meltzer, Replication of rs300774, a genetic biomarker near ACP1, associated with suicide attempts in patients with schizophrenia: Relation to brain cholesterol biosynthesis, J Psychiatr Res 94 (2017) 54-61.

Reviewer 2 Report

This a very well written and comprehensive review on the role of RhoA pathway in schizophrenia and its potential as a treatment target. The review covered the schizophrenia risk genes in RhoA pathway, associated rodent models and druggable sites in the signaling pathway. The review is well structured; each section is easy to follow and provides detailed information; figure and tables are clear and concise. I do not have any concern.

Author Response

The authors would like to thank the reviewers and editors for their time and effort in reviewing this manuscript. 

Reviewer 3 Report

In this manuscript, Rinako Tanaka and Kiyofumi Yamada review the literature on RhoS GTPase and its association with schizophrenia. They describe the basic regulation mechanisms of Rho GTPases and then review the evidence from genomic studies that supports the implication of RhoA regulator or effector pathways in schizophrenia. They also describe the available preclinical mouse models and the data regarding the use of RhoA pathway inhibitors in drug-induced schizophrenia animal models.

Overall this is a very well organized and written review paper. I have only three minor comments.

Comment 1. Introduction lines 58-60. Here the authors should update the information on SCZ-associated loci by citing the latest Nature paper (https://www.nature.com/articles/s41586-022-04434-5)

Comment 2. Section 6, lines 380-390. This paragraph needs restructuring and rephrasing. The analogy with EGFR-targeted anticancer drugs is not clear. It may actually suggest the opposite of what the authors try to put forward, I think. 

Comment 3.  Table 3. The MK801 experinents concerning locomotion and social interaction apparently belong to the glutamate group, no? It is not clear from the formatting of the Table.

Author Response

The authors would like to thank the reviewers and editors for their time and effort in reviewing this manuscript. Please find the detailed responses below and the corresponding revisions highlighted changes in the re-submitted files.

Comment 1) Introduction lines 58-60. Here the authors should update the information on SCZ-associated loci by citing the latest Nature paper (https://www.nature.com/articles/s41586-022-04434-5)

 (Answer) Thank you for your valuable comment. We have updated the information on SCZ-associated loci by citing the latest Nature paper (https://www.nature.com/articles/s41586-022-04434-5). (Page 2, lines 59-62)

Comment 2) Section 6, lines 380-390. This paragraph needs restructuring and rephrasing. The analogy with EGFR-targeted anticancer drugs is not clear. It may actually suggest the opposite of what the authors try to put forward, I think. 

(Answer) We have restructured and rephrased using the example of drug development of pain therapy. This case is same as what we try to put forward (Page 9, line 389-402).

Comment 3) Table 3. The MK801 experiments concerning locomotion and social interaction apparently belong to the glutamate group, no? It is not clear from the formatting of the Table.

(Answer) Yes, it is. We have revised Table 3 to easily find out for the reader which model belongs to the respective phenotype (Page 10).